# Psychological stress disturbs bone metabolism via miR-335-3p/Fos signaling in osteoclast

Jiayao Zhang[1], Juan Li[1], Jiehong Huang[2], Xuerui Xiang[2], Ruoyu Li[2], Yun Zhai[1], Shuxian Lin[1]*†, Weicai Liu[1]*†

[1]Shanghai Engineering Research Center of Tooth Restoration and Regeneration & Tongji Research Institute of Stomatology & Department of Prosthodontics, Shanghai Tongji Stomatological Hospital and Dental School, Tongji University, Shanghai, China; [2]Department of Neurology and Neurological Rehabilitation, Shanghai Disabled Persons' Federation Key Laboratory of Intelligent Rehabilitation Assistive Devices and Technologies, Yangzhi Rehabilitation Hospital (Shanghai Sunshine Rehabilitation Center), School of Medicine, Tongji University, Shanghai, China

**\*For correspondence:**
shuxian.lin@hotmail.com (SL);
vogi@163.com (WL)

†These authors contributed equally to this work

**Competing interest:** The authors declare that no competing interests exist.

## eLife Assessment

The article presents **valuable** findings of bone remodeling under chronic unpredictable mild stress (CUMS). This is an interesting work on mental stress on bone health and osteoporosis, and the authors offer **solid** evidence of decreased bone mass mediated by miR-335-3p/Fos signaling in osteoclasts that are involved in the induction of bone loss caused by CUMS. This revised version provides new data that improved the quality of the article and addressed the reviewers' concerns.

**Abstract** It has been well validated that chronic psychological stress leads to bone loss, but the underlying mechanism remains unclarified. In this study, we established and analyzed the chronic unpredictable mild stress (CUMS) mice to investigate the miRNA-related pathogenic mechanism involved in psychological stress-induced osteoporosis. Our result found that these CUMS mice exhibited osteoporosis phenotype that is mainly attributed to the abnormal activities of osteoclasts. Subsequently, miRNA sequencing and other analysis showed that miR-335-3p, which is normally highly expressed in the brain, was significantly downregulated in the nucleus ambiguous, serum, and bone of the CUMS mice. Additionally, in vitro studies detected that miR-335-3p is important for osteoclast differentiation, with its direct targeting site in *Fos*. Further studies demonstrated FOS was upregulated in CUMS osteoclast, and the inhibition of FOS suppressed the accelerated osteoclastic differentiation, as well as the expression of osteoclastic genes, such as *Nfatc1*, *Acp5*, and *Mmp9*, in miR-335-3p-restrained osteoclasts. In conclusion, this work indicated that psychological stress may downregulate the miR-335-3p expression, which resulted in the accumulation of FOS and the upregulation of NFACT1 signaling pathway in osteoclasts, leading to its accelerated differentiation and abnormal activity. These results decipher a previously unrecognized paradigm that miRNA can act as a link between psychological stress and bone metabolism.

## Introduction

Nowadays, chronic psychological stress, a critical public health issue, is getting more and more attention. It is responsible for various systemic diseases, one of which is the disruption of bone metabolism.

Several studies had shown that chronic stress and even depression can lead to osteoporosis (*Calarge et al., 2014*; *Furlan et al., 2005*; *Yirmiya et al., 2006*). Whereas how does psychological stress signal the brain but affect peripheral organs? Former research suggested that psychological stress can stimulate the autonomic nerves or the central nervous system to disrupt the activation of sympathetic nerves (*Kondo and Togari, 2003*; *Ma et al., 2024*) and endocrine homeostasis (*Raison and Miller, 2003*), respectively. They alter the levels of neurotransmitters (*Chen and Wang, 2017*; *Foertsch et al., 2017*; *Guo et al., 2023*; *Khosla et al., 2018*; *Teong et al., 2017*), neuropeptides (*Baldock et al., 2014*; *Ng and Chin, 2021*), hormones (*Hasan et al., 2012*; *Henneicke et al., 2017*), related biomolecules, and their corresponding receptors in vivo, so as to affect the activations of several signaling pathways. Therefore, it contributes to increased osteoclast activity, decreased osteoblast activity, and promoted bone matrix degradation, ultimately leading to bone loss. However, treatment of antidepressant and molecules targeting the above-mentioned receptors can only partially alleviate bone loss (*Lam et al., 2022*; *Marenzana et al., 2007*), which suggest that there may be other unanticipated molecular mechanisms involved in psychological stress-induced osteoporosis and efficient for the clinical treatment. Besides, these mentioned neurotransmitters or hormones, and their related receptors have broad biological effects. Their effects in vivo are highly dose-dependent, which usually give rise to unpredictable side effects when using hormonotherapy or neuropeptide therapy for patients with psychological stress-related osteoporosis (*Kim et al., 2021b*; *Marenzana et al., 2007*; *Teong et al., 2017*). Therefore, it is full of necessity to further explore alternative mechanisms that are implicated in psychological stress-related osteoporosis since any of these new discoveries will be possible for attracting an ideal clinical therapy with higher biosafety and efficiency.

MicroRNAs (miRNAs) carried by extracellular vesicles (EVs) or proteins in vivo are highly involved in biocommunication between multiple organs. They usually have a high regulatory efficiency for targeting multiple transcripts, and a wide regulatory range for controlling the expression of genes coding for protein. Particularly, due to the protection of EVs, they are not only stable in the extracellular environment, such as blood (*Chen et al., 2008*), but also free to pass through the blood–brain barrier, indicating that miRNAs may work as the linkers between brain and bone. Consistently, a previous study reported that miR-483-5p, stored in brain-derived EVs, could promote bone–fat imbalance in Alzheimer's disease (*Liu et al., 2023*). Taking the consideration that several endogenic miRNAs in the brain (*Olejniczak et al., 2018*) and blood (*Makrygianni and Chrousos, 2023*) were reported to change their expression levels after chronic psychological stress, it is deducible that psychological stress may interfere the bone metabolism through brain-derived miRNAs. Therefore, it is significant and valuable to explore the biofunctions of brain-derived miRNAs during the pathogenesis of psychological stress-related osteoporosis, which can also preliminarily discover the roles of brain-derived miRNAs during the central–peripheral linkage. What's more, as miRNAs are ideal targets for precision therapy due to their effective biofunctions, easy synthesis, and longer half-life, exploring their regulatory role in the pathogenesis of the psychological stress-related osteoporosis is crucial for building the theoretical foundation, based on which a more efficient and much safer clinical treatment can be developed.

In this context, we primarily established the chronic unpredictable mild stress (CUMS) model mice, characterized its osteoporosis phenotype, and analyzed the activities of osteoblasts and osteoclasts. Subsequently, using high-throughput miRNA-seq analysis of bone, we identified the potential miRNA candidate that is responsible for bone loss under psychological stress, as well as investigated the molecular mechanism of this miRNA. Our findings show that the abnormal osteoclast activity is much more responsible for the psychological stress-induced osteoporosis. Moreover, miR-335-3p, which can inhibit *Fos* translation and its integration with *Nftac1*, is significantly reduced under psychological stress and sequentially enhances the activation of NFATC1 signaling pathway, thereby upregulating the osteoclast differentiation and maturation. These results imply that miR-335-3p plays an important role in the communication between the central nervous system and peripheral bone tissue; in addition, it provides a new perspective on neuroregulatory mechanisms of the brain that participated in bone metabolism when suffering psychological stress.

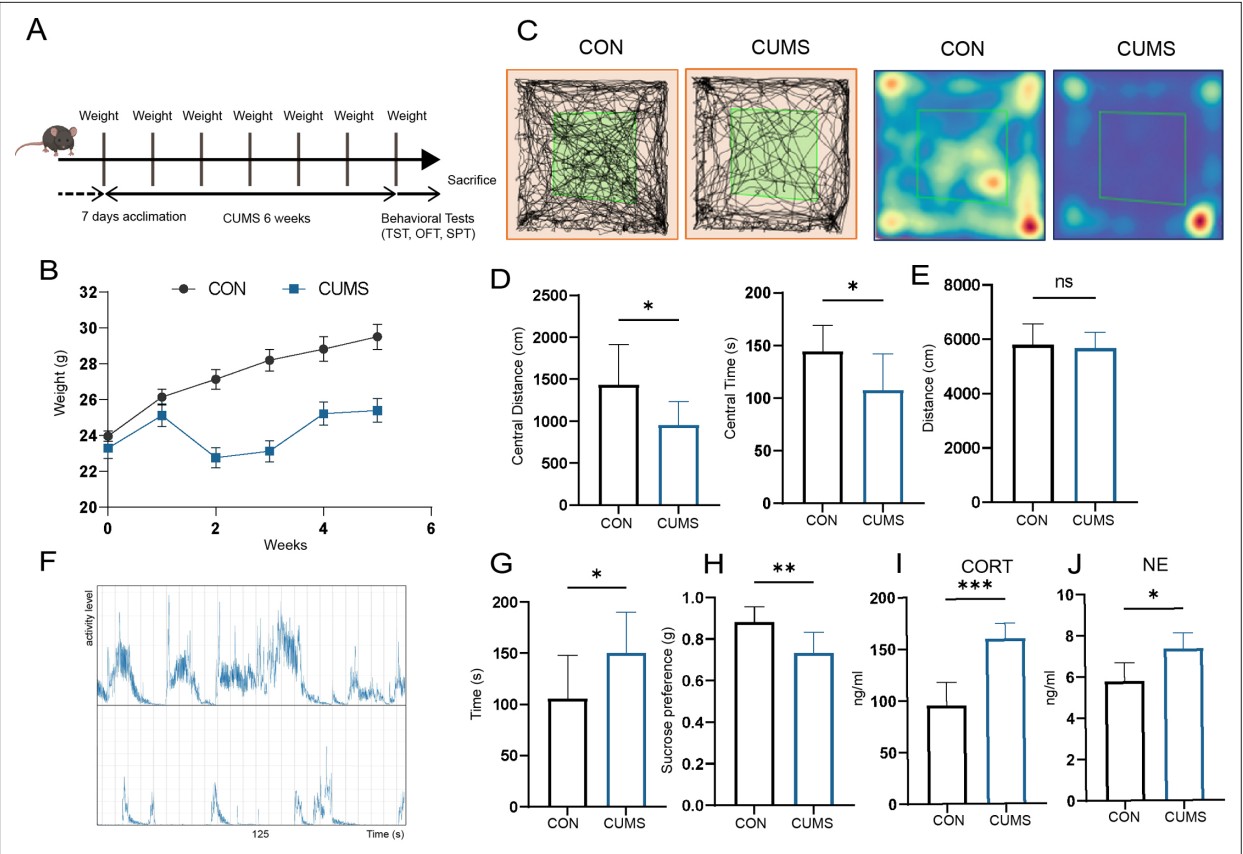

**Figure 1.** Effect of chronic unpredictable mild stress (CUMS) on body weight, behavioral performance, and serum metabolism. (**A**) Schematic of the procedure used to establish a CUMS model in mice. (**B**) Body weight (n = 9). (**C**) Distance moved and (**D**) time spent in the central area of open-field test (OFT) (n = 9). (**E**) Total distance moved in OFT (n = 9). (**F**) Frequency and amplitude of body movements during struggle. (**G**) Immobility in tail suspension test (TST) (n = 9). (**H**) Sucrose preference index (n = 9). (**I**) Serum corticosterone (CORT) levels (n = 6). (**J**) Serum norepinephrine (NE) levels (n = 5). All data are presented as means ± SD; ns, p>0.05, *p<0.05, **p<0.01, ***p<0.001, by unpaired Student's *t*-test.

The online version of this article includes the following source data for figure 1:

**Source data 1.** Full dataset for *Figure 1*.

## Results

### CUMS induces behavioral and physiological changes

The mental state of mice was examined by body weight, behavioral and serum parameters. It is shown that the CUMS mice had significantly lower body weights (*Figure 1B*) compared to the control group. They had considerably lower distances (*Figure 1C*) and times (*Figure 1D*) within the central area of the open field but did not differ from the control group in terms of the total distance moved in the open field (*Figure 1E*). Besides, the tail-hanging experiment showed that the immobilization time of mice in the CUMS group to give up struggling was significantly increased (*Figure 1F and G*). At the same time, they had a lower sucrose preference index (*Figure 1H*). In addition, serum corticosterone (CORT) (*Figure 1J*) and norepinephrine (NE) (*Figure 1J*) levels were significantly elevated in CUMS mice. Taken together, these results all indicated that CUMS modeling caused the mice to develop anxiety and depression-like behaviors.

### CUMS leads to bone loss

Next, mouse femurs from different groups were dissected for micro-CT scanning (*Figure 2A–D*) and H&E staining (*Figure 2E*). Then, quantitatively analyzed histomorphometry, bone mineral density, and bone microarchitecture to assess the bone remodeling were influenced by psychological stress. It was found that the femurs of CUMS mice had sparse cancellous bone and thinner cortical bone compared to control mice, indicating that psychological stress harmed both cancellous and cortical bone.

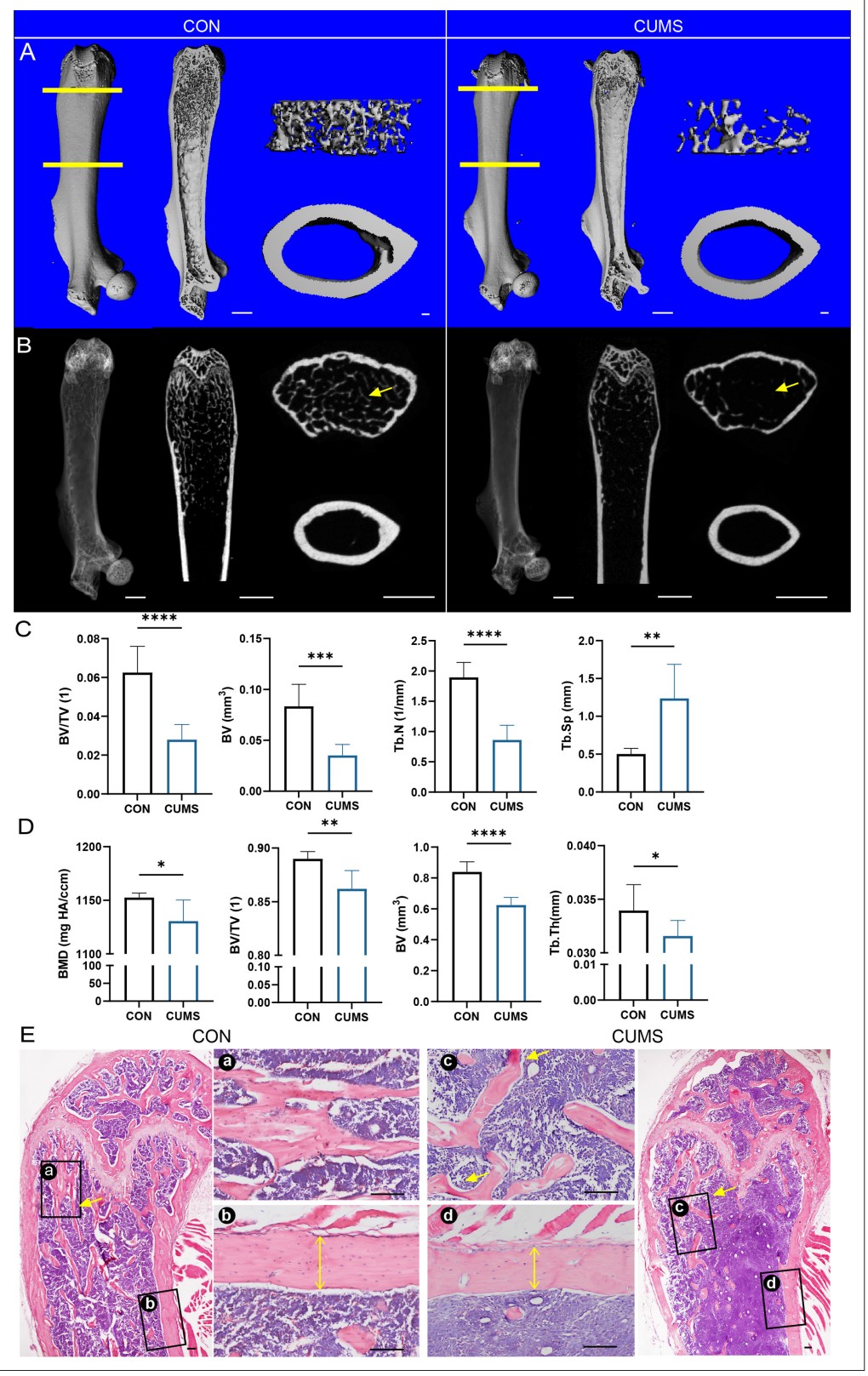

**Figure 2.** Bone mass of the femur decreased in the chronic unpredictable mild stress (CUMS) model. (**A**) Three-dimensional reconstruction images of femurs from control and CUMS mice. (**B**) Representative X-ray images of femurs from control and CUMS mice. Quantitative micro-CT analysis of femur (**C**) trabecular bone and (**D**) cortical

*Figure 2 continued on next page*

*Figure 2 continued*

bone. (**E**) H&E staining, scale bar, 200 um. All data are presented as mean ± SD, n = 7; *p<0.05, **p<0.01, ***p<0.001, by unpaired Student's *t*-test.

The online version of this article includes the following source data for figure 2:

**Source data 1.** Full dataset for *Figure 2*.

## CUMS promotes bone formation slightly but enhances osteoclast activity significantly

To further explore the cause of psychological stress-induced bone loss, the osteogenic activity of mice was then examined in the CUMS group. Firstly, Masson staining visualized that more osteoid was present in cortical and cancellous bone in CUMS mice (*Figure 3A*). In addition, the distance between two fluorochromes calcein lines was found wider in CUMS mice, suggesting the mineral deposition rate in CUMS mice was faster than control ones (*Figure 3D*). Then RNA were isolated to examine osteoblast-related genes and found that the expression of osterix (*Osx*), osteocalcin (*Ocn*), osteopontin (*Opn*), and dentin matrix protein 1 (*Dmp1*) was elevated at the mRNA level in CUMS mice, whereas no significant difference was observed in runt-related transcription factor 2 (*Runx2*) and alkaline phosphatase (*Alpl*) (*Figure 3E*). Besides, immunohistochemical staining of OSX and OCN displayed that the protein expression of OSX was higher in CUMS mice (*Figure 3B*), but no considerable difference could be found in OCN (*Figure 3C*).

Since the enhanced osteogenesis could not explain the reduced bone mass under psychological stress, we further examined the osteoclastic activity of CUMS mice. Tartrate-resistant acid phosphatase (TRAP) staining showed a significant increase in the number and area of TRAP-positive osteoclasts under CUMS (*Figure 3G and H*). Then, RT-PCR analysis of mouse femurs revealed that several osteoclastic genes, such as *Ca2*, *Mmp9*, *Nfatc1*, and *Acp5*, were significantly increased in CUMS mice (*Figure 3I*, *Figure 3—figure supplement 2*). Consistently, immunohistochemistry analyses also showed an elevated MMP9 expression in CUMS mice (*Figure 3J*). What's more, the serum concentration of TRAP, calcium (CA), and phosphate (P) also increased in CUMS mice, while parathyroid hormone (PTH) level in serum decreased (*Figure 3—figure supplement 1*), which indicated that they might be in the initial phase of osteoporosis. The enhanced osteoclastic function promoted ions release from the bone matrix and in turn slightly promote the level of CA and P and suppress the PTH secretion temporarily. Thus, these data demonstrate that osteoclastic activity is elevated in vivo under psychological stress, which may be a more direct and vital cause of the decline in bone mass.

## miR-335-3p is apparently downregulated in CUMS mice

To search the key miRNAs involved in the abnormal bone metabolism correlated to chronic stress, femurs (dissected off the growth plate) were collected from different groups for sRNA-seq, respectively. The heat map and volcano plot analysis displayed that a number of miRNAs were significantly altered in CUMS mice (*Figure 4A and B*). The most significant variations including miR-335-3p, miR-133a-3p, miR-1298-5p, miR-144-5p, miR-1b-5p, mi5-582-3p, based on the criteria of |Log2 Fold Change (FC)|>1.5.

In addition, based on these differentially expressed miRNAs, miRDB and miRTarBase databases were used to predict potential target genes, which then performed bioinformatics analysis. In total, 184 Gene Ontology (GO) terms were enriched in target genes according to the criteria false discovery rate (FDR) < 0.05. The results identified alteration in the DNA binding, transcription activity, cell differentiation, and nervous system development, with most predicted genes being enriched (*Figure 4C and D*).

Later, a 3.4-fold and 31-fold decrease in miR-335-3p, miR-133a-3p was further verified by qRT-PCR (*Figure 4E*). Since upregulation of miR133a has been previously reported to play an important role in postpartum osteoporosis (*Li et al., 2018*), its function seems to be opposite to that in CUMS mice. Together, these results suggest that the decline of miR-335-3p in CUMS mice may be of more concern to us.

In addition, it is stress signals received by the brain that induced local miR-335-3p alterations in bone tissue. miR-335-3p is highly expressed in the brain, according to the mouse small noncoding RNA tissue atlas (https://www.ccb.uni-saarland.de/asra/; *Figure 4—figure supplement 1*). So the

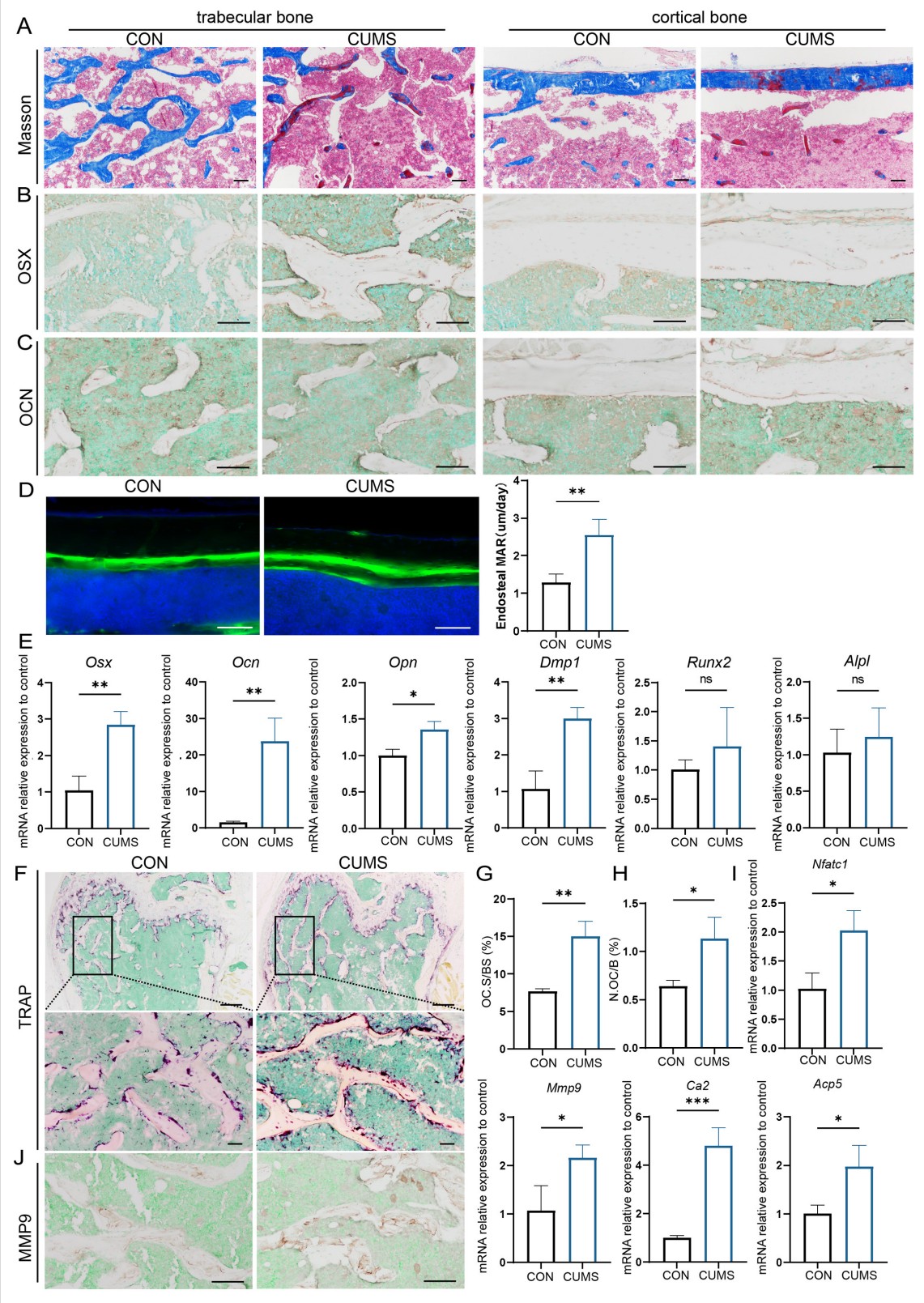

**Figure 3.** Osteogenic and osteoclastic metabolism of mouse femur under psychological stress. (**A**) Masson staining. (**B, C**) Representative IHC images for OSX, OCN expression in the femur. (**D**) Calcein double labeling of cortical bone with quantification of MAR. (**E**) qRT-PCR quantification analysis of osteogenic markers including *Osx*, *Ocn*, *Opn*, *Dmp1*, *Runx2*, and *Alpl*. (**F**) Representative tartrate-resistant acid phosphatase (TRAP) staining images of the femur with quantitative analysis of (**G**) osteoclast surface per bone surface (Oc.S/BS) and (**H**) number per bone surface (N.OC/B). (**I**) qRT-PCR

*Figure 3 continued on next page*

*Figure 3 continued*

quantification analysis of osteogenic markers including *Ca2*, *Mmp9*, *Nfatc1*, and *Acp5*. (**J**) Representative IHC images for MMP9 in the femur. All data are presented as mean ± SD, n = 3; *p<0.05, **p<0.01, ***p<0.001, by unpaired Student's *t*-test. Scale bar, 100 um.

The online version of this article includes the following source data and figure supplement(s) for figure 3:

**Source data 1.** Full dataset for *Figure 3*.

**Figure supplement 1.** Detection of osteogenic and osteoclastic markers in serum.

**Figure supplement 1—source data 1.** Full dataset for *Figure 3—figure supplement 1*.

**Figure supplement 2.** qRT-PCR quantification analysis of osteoclastic markers, including *Oscar*, *Dastamp*, *Calar*, *Ctsk*, *Rank/Opg*, and *Clcn7*.

**Figure supplement 2—source data 1.** Full dataset for *Figure 3—figure supplement 2*.

expression of miR-335-3p were detected subsequently in five critical brain regions, including medial prefrontal cortex (mPFC), nucleus ambiguous (NAC), amygdala, hypothalamus, and hippocampus, which play key roles in receiving and processing chronic stress. Only miR-335-3p was significantly decreased in NAC (*Figure 4G*). Consistently, the miR-335-3p level in the serum of CUMS mice was also reduced (*Figure 4H*). These results imply that brain-derived and femur-expressed miR-335-3p maintain bone homeostasis under normal conditions. Psychological stress leads to a decrease in miR-335-3p secreted by NAC, which in turn reduces levels in the femur.

## The decrease in miR-335-3p promotes osteoclast activity and bone loss

To further confirm whether miR-335-3p affected osteoclast activity, RAW264.7 cells were transfected with mimic-NC, mimic-miR-335-3p, inhibitor-NC, or inhibitor-miR-335-3p, respectively (*Figure 5—figure supplement 1*). Cytochemical TRAP staining (*Figure 5A–C*) and pit formation assay (*Figure 5D and E*) showed that mimic-miR-335-3p impaired the activity and function of osteoclasts, while inhibitor-miR-335-3p significantly promotes the activity and function of osteoclasts. Next, the mRNA levels of *Acp5*, *Nfatc1*, and *Mmp9* in the indicated RAW264.7 cells were determined using qRT-PCR. A significantly decreased expression in these osteoclast-related genes was observed in the mimic-miR-335-3p group, but an increased expression happened in the inhibitor-miR-335-3p group (*Figure 5F*). Additionally, in the inhibitor-miR-335-3p group, more intranuclear NFATC1, FOS, and co-localization signals were observed by immunofluorescence staining, and this response was inhibited by transfection of mimic (*Figure 5G*). More importantly, mice injected with antagomir-miR-335-3p showed significant bone loss, although the bone mass in mice injected with agomir-miR-335-3p did not find a remarkable change (*Figure 5I*). These results suggest that miR-335-3p may play an important role in vivo as a bone protector.

## miR-335-3p alters osteoclast activity by targeting Fos

To seek the downstream molecular mechanisms that mediate the effects of miR-335-3p on the osteoclast, the ingenuity pathways analysis was made based on miR-335-3p and its predicted target gene in the miRDB and miRTarBase databases. The results show that *Fos* is one of the mRNA that miR-335-3p could target with a high prediction score and have a tight connection with osteoclast differentiation and maturation and osteoporosis (*Figure 6A*). FOS has been previously described as a regulator of transcription factors. It could induce NFATC1 translocation and finally increase the expression of osteogenic markers such as *Acp5*, *Ctsk*, and *Mmp9* (*Deng et al., 2022*). Kyoto Encyclopedia of Genes and Genomes (KEGG) analysis based on the above predicted target genes resulted in the same higher enrichment for transcription factors activity and DNA binding (*Figure 6B*).

Next, the dual-luciferase, co-transfection of miR-335-3p, suppressed the luciferase activity of the reporter containing wildtype *Fos* 3'UTR sequence by dual-luciferase reporter assay, which further strongly verifies the direct target relationship between miR-335-3p and *Fos*. But it failed to completely return to normal luciferase activity of two predicted target sites (*Figure 6D*) deleted construct, which means other binding sites still exist (*Figure 6C*). Furthermore, at the protein level, there were stronger signals that FOS co-localized with CTSK in the femur of CUMS mice, whose miR-335-3p level decreased (*Figure 6E*), although there was no significant difference at the mRNA level (*Figure 6—figure supplement 1*). These results suggest that the enhanced osteoclastic activity under psychological stress may be due to the reduction in miR-335-3p in the femur, which in turn slows down its inhibitory effect on *Fos* in osteoclast.

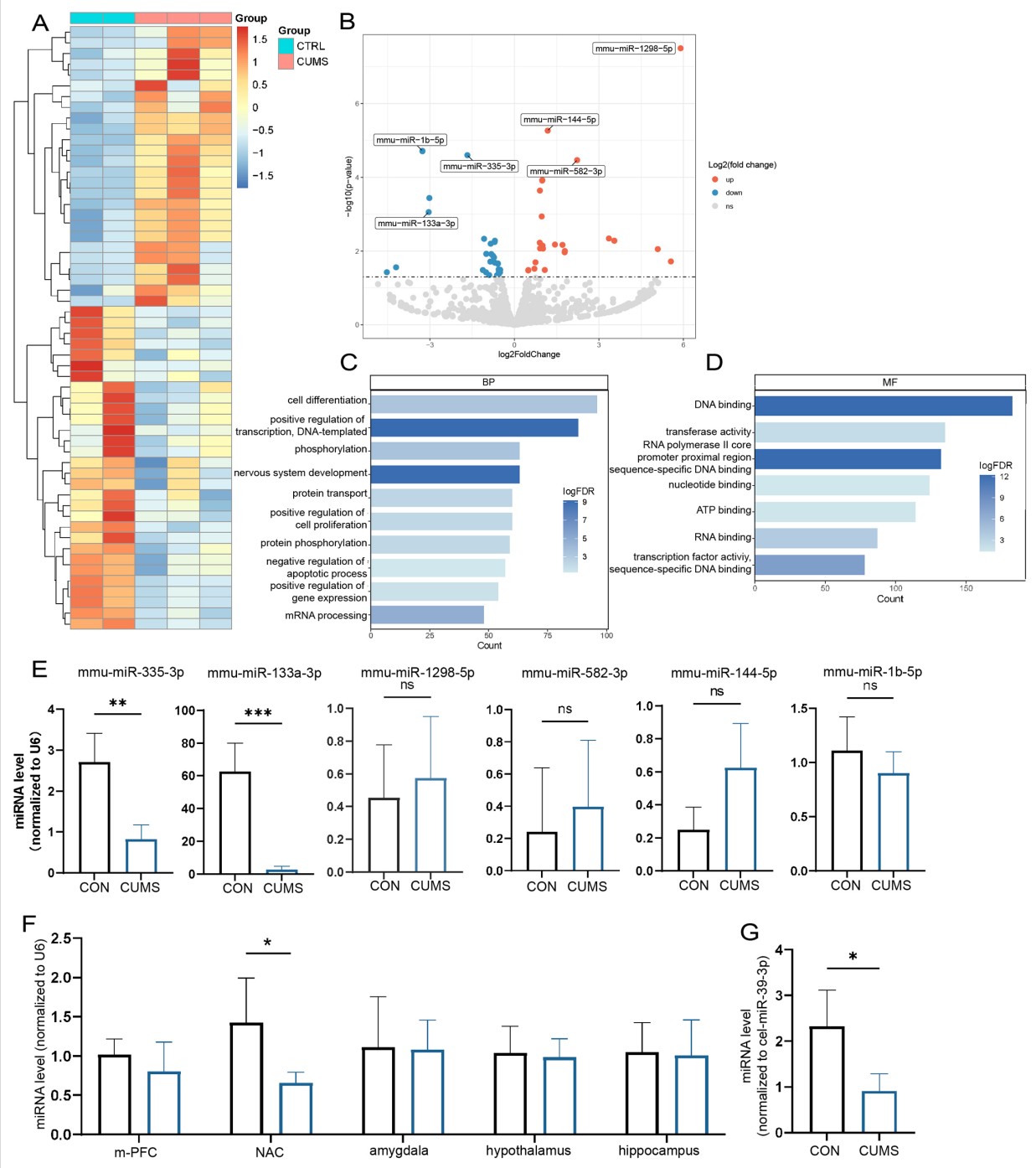

**Figure 4.** Screening of key miRNAs. (**A**) The heatmap and (**B**) volcano plot for visualization of differentially expressed miRNAs in the femur proximal to the distal femoral growth plate (p<0.05). (**C, D**) Gene Ontology (GO) terms associated with targeted mRNA of differentially expressed miRNAs between groups obtained by database (miRDB and miRTarBase) prediction. (**E**) Validation by real-time expression analysis of miRNAs with statistically significant differences (Con, n = 3; chronic unpredictable mild stress [CUMS], n = 5). (**F**) Real-time expression of miR-335-3p in five stress-related brain regions (n = 4) and (**G**) serum (n = 3). All data are presented as mean ± SD, n = 5; *p<0.05, **p<0.01, ***p<0.001, by unpaired Student's *t*-test.

The online version of this article includes the following source data and figure supplement(s) for figure 4:

**Source data 1.** Full dataset for *Figure 4A–D*.

**Source data 2.** Full dataset for *Figure 4E–G*.

**Figure supplement 1.** Reads per million mapped reads (RPMM) expression of miR-335-3p in different tissues.

**Figure supplement 1—source data 1.** Full dataset for *Figure 4—figure supplement 1*.

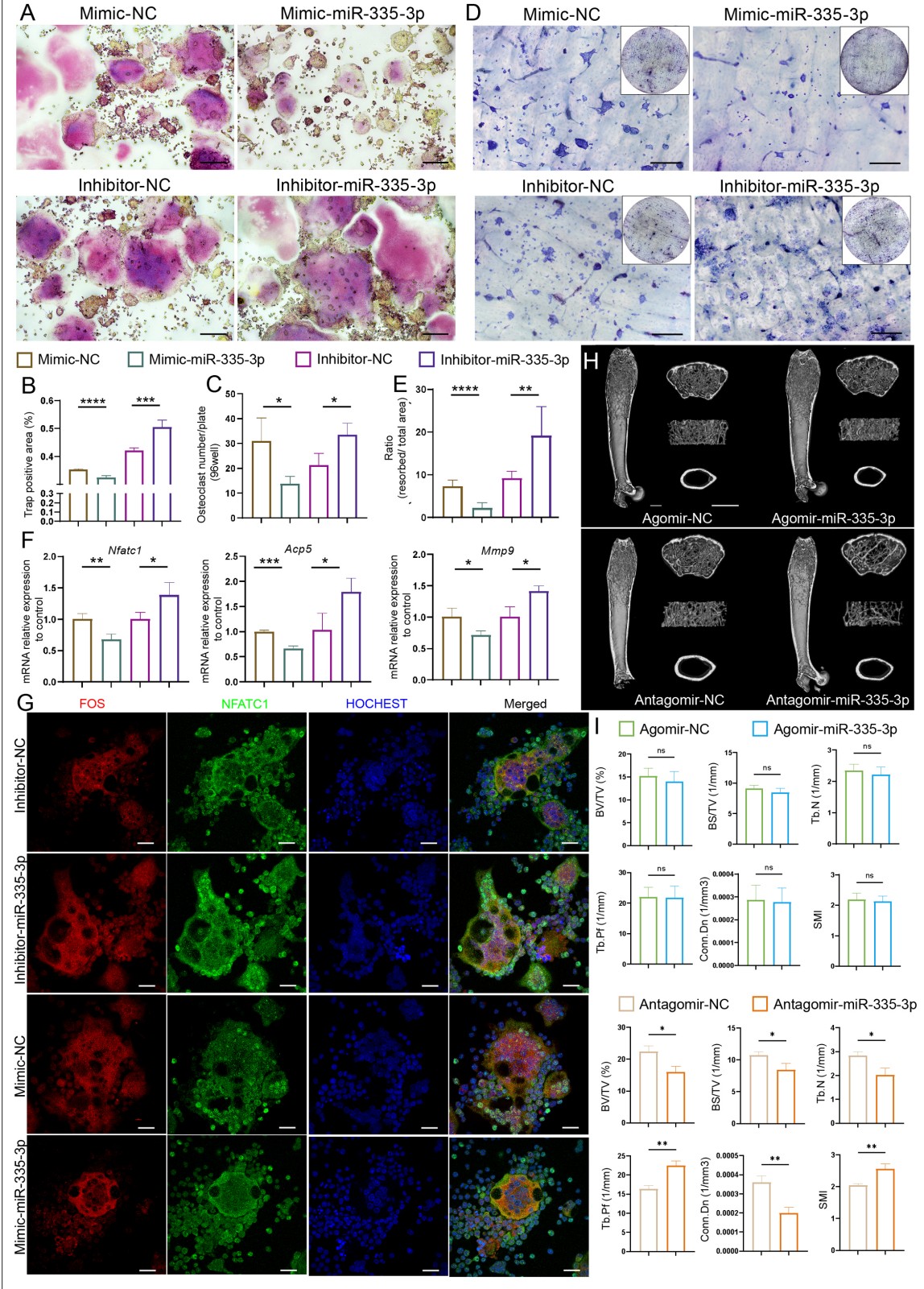

**Figure 5.** Initial validation of the function of miR-335-3p. (**A**) Representative cytochemical tartrate-resistant acid phosphatase (TRAP) staining images, Scale bar, 200 um. (**B**) The amounts of osteoclasts (multinucleated TRAP-positive cells), n = 5. (**C**) The positive TRAP area of osteoclasts, n = 5. (**D**) Representative resorption pits, Scale bar, 200 um. (**E**) The area of resorbed surface, n = 6. (**F**) qRT-PCR quantification analysis of *Acp5*, *Nfatc1*, *Mmp9* expression of RANKL-induced RAW264.7 transfected with mimic-NC, mimic-miR-335-3p, inhibitor-NC, or inhibitor-miR-335-3p, n = 3. (**G**) Representative

*Figure 5 continued on next page*

*Figure 5 continued*

IF images for FOS, NFATC1, and HOCHEST expression, Scale bar, 200 um. (**H**) Three-dimensional reconstruction images of femurs from agomir NC, agomir-mir-335-3p, antagomir NC, and antagomir-miR-335-3p mice. (**I**) Quantitative micro-CT analysis of femur trabecular bone, n = 3; scale bar, 1 mm. All data are presented as mean ± SD; *p<0.05, **p<0.01, ***p<0.001, by unpaired Student's *t*-test.

The online version of this article includes the following source data and figure supplement(s) for figure 5:

**Figure supplement 1.** Cell transfection efficiency.

**Source data 1.** Full dataset for *Figure 5*.

**Figure supplement 1—source data 1.** Full dataset for *Figure 5—figure supplement 1*.

Moreover, the effect of promoting osteoclastic activity of inhibitor-miR-335-3p could be reversed via T-5224, an inhibitor of FOS, measured via TRAP staining and counting (*Figure 6F–H*). The opposing effects on the mRNA levels mentioned above were also exerted via T-5224 (*Figure 6I*). These results further demonstrated that miR-335-3p regulates osteoclast activity by targeting Fos.

## Discussion

Here, we found that psychological stress disturbs bone metabolism in vivo and ultimately leads to bone loss in CUMS mice. We reported for the first time that miR-335-3p could target *Fos* to interfere with its translation and inhibit the activity of NFATC1 signaling pathway. Besides, it was significantly reduced in mouse NAC, serum, and bone tissues under psychological stress. Therefore, the osteoclast activity is apparently promoted in CUMS mice, leading to psychological stress-induced osteoporosis. Our study demonstrated that miR-335-3p is a key mediator for nerve-regulated bone metabolism under psychological stress, and it might hold great potential to be a novel therapeutic target (*Figure 7*).

Previous studies have attributed psychological stress-induced osteoporosis to the altered neurotransmitters, neuropeptides, and hormones that circulate and localize in tissues, such as corticosterone, noradrenaline, serotonin, and sex hormones (*Chrousos, 2009*; *Wang et al., 2023*; *Zhao et al., 2024*). The prevailing opinion explaining this phenomenon is that the brain receives signals from the stressor, subsequently passing the information through the hypothalamic-pituitary-adrenal cortex axis, the locus ceruleus-sympathetic-adrenal medulla system (*Miller and O'Callaghan, 2002*), and other intermediary pathways. Ultimately, these stimulate the peripheral endocrine glands and nerve endings, which in turn results in the aforementioned alterations. However, apart from the sensitive and potent regulation capacity of our body, are there more direct pathways involved in the regulation of bone metabolism under psychological stress? In recent years, there has been an increasing focus on the regulatory role of miRNAs in the interaction between organs or systems since miRNAs protected by EV can be stably stored and transported to nearby or even isolated organs and systems in vivo. Especially their ability to overcome the blood–brain barrier and broaden new horizons for the studies on the interaction between the central nervous system and peripheral tissues, for example, bone. Currently, it has been reported that the expression profile of miRNAs was significantly altered in patients with neuropsychiatric disorders (*Rao et al., 2013*; *Wang, 2021*; *Wu et al., 2022*; *Żurawek and Turecki, 2021*). Some of these miRNAs contained in EVs play a vital role in the central regulation of bone metabolism. miR-328a-3p and miR-150-5p enriched in EVs from the hippocampus after traumatic brain injury accelerated bone healing (*Xia et al., 2021*). miR-384-5p in brain-derived EVs induces a bone–fat imbalance and bone loss in Alzheimer's disease (*Liu et al., 2023*). Psychological stress, one of the most common types of neuropsychiatric disorders, is well established as a promoter of osteoporosis, so could miRNAs act as messengers between it and bone homeostasis?

Psychological stress is correlated to several brain regions, such as the mPFC, the NAC, the amygdala, the hypothalamus, and the hippocampus. Expression levels of several miRNAs related to bone metabolism were altered in the brain under psychological stress. Hypotheses have been developed on miRNAs that may regulate bone loss under chronic stress (*He et al., 2021*). However, no study has yet reported whether altered miRNAs in the brain under stress can influence bone metabolism through blood circulation. In this study, for the identification of the key brain region involved in chronic psychological stress-related bone metabolism, several miRNAs, which changed their expression levels in the CUMS femur compared to the control ones, were examined in the abovementioned brain

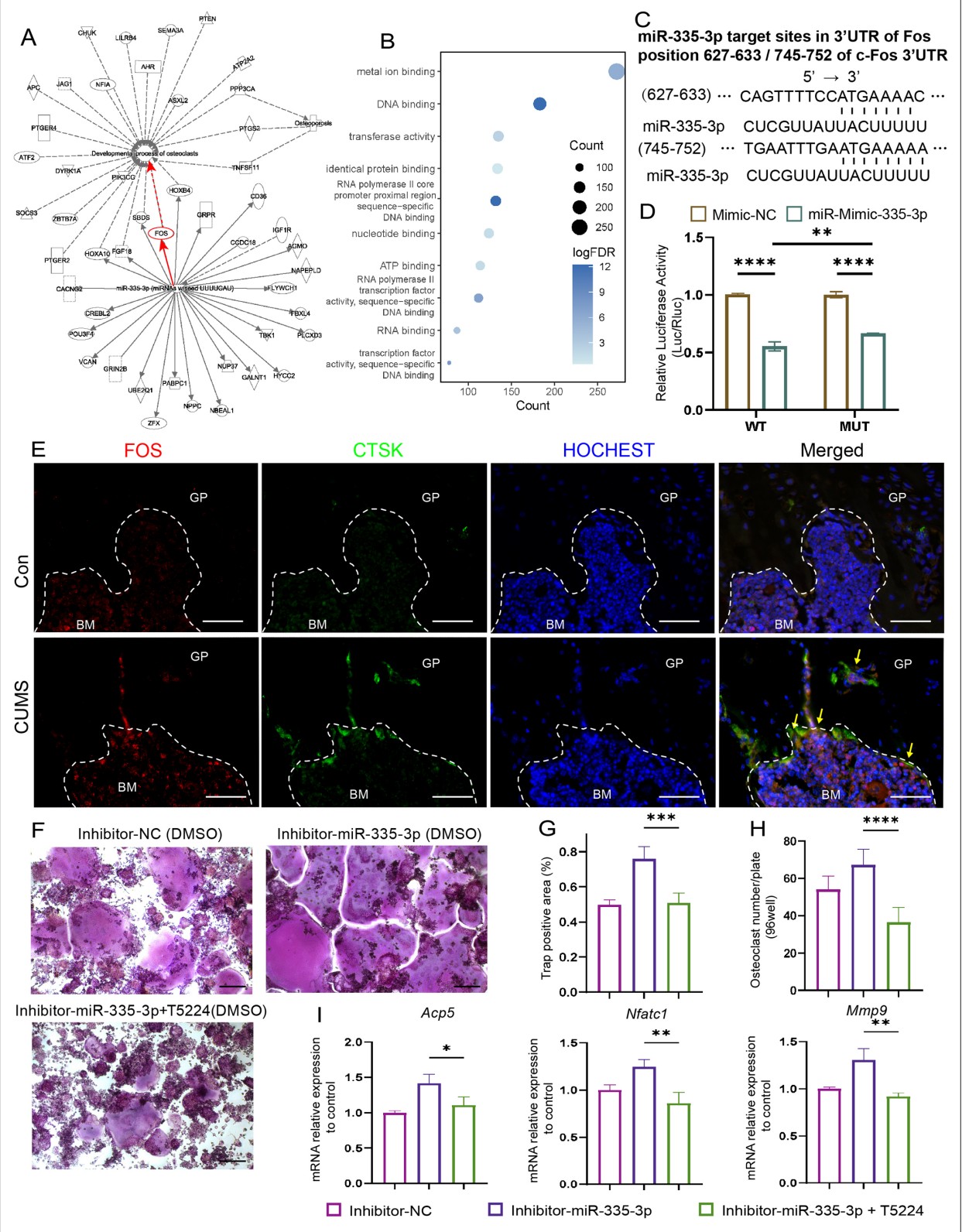

**Figure 6.** Seek and verification mechanism of miR-335-3p in vivo and in vitro. (**A**) The ingenuity pathways analysis (IPA) of miR-335-3p and its predictive target gene. (**B**) Kyoto Encyclopedia of Genes and Genomes (KEGG) terms associated with targeted mRNA of differentially expressed miRNAs between groups obtained by database (miRDB and miRTarBase) prediction. (**D**) Firefly luciferase activity (n = 3). (**C**) Sequence alignment of miR-335-3p and its predictive target sites in 3'UTR of *Fos*. (**E**) Representative IF images for FOS, CTSK, and HOCHEST expression in the femur (scale bar, 50 um).

*Figure 6 continued on next page*

*Figure 6 continued*

(**F**) Representative cytochemical tartrate-resistant acid phosphatase (TRAP) staining images (scale bar, 200 um). (**G**) The area and (**H**) the number of osteoclasts (multinucleated TRAP-positive cells) (n = 4). (**I**) qRT-PCR quantification analysis of *Acp5*, *Nfatc1*, *Mmp9* expression of RANKL-induced RAW264.7 transfected with inhibitor-NC, miR-335-3p-inhibitor, or miR-335-3p-inhibitor+T5224 (n = 3). All data are presented as mean ± SD; *p<0.05, **p<0.01, ***p<0.001, ****p<0.0001, by one-way ANOVA with Tukey's post hoc test.

The online version of this article includes the following source data and figure supplement(s) for figure 6:

**Source data 1.** Full dataset for *Figure 6*.

**Figure supplement 1.** qRT-PCR quantification analysis of *Fos* expression in femur of control and chronic unpredictable mild stress (CUMS) mice.

**Figure supplement 1—source data 1.** Full dataset for *Figure 6—figure supplement 1*.

regions, such as the mPFC, NAC, amygdala, and hypothalamus. It was found that miR-335-3p, which is originally highly expressed in the brain and closely associated with bone metabolism, was significantly downregulated in the NAC, as well as the serum of CUMS mice. This result is in accordance with the miRNAs sequencing results in the NAC of the CUMS mice reported by *Ma et al., 2019*. Taken together, it indicated that the NAC might be one of the important brain regions that is responsible for the psychological stress-induced osteoporosis via its secretion of miR-335-3p. In other words, these also suggest that the NAC-derived and blood-transported miR-335-3p may have a protective effect on bone homeostasis under normal conditions, while in contrast, chronic stress diminishes the protective effect of miR-335-3p and leads to bone loss.

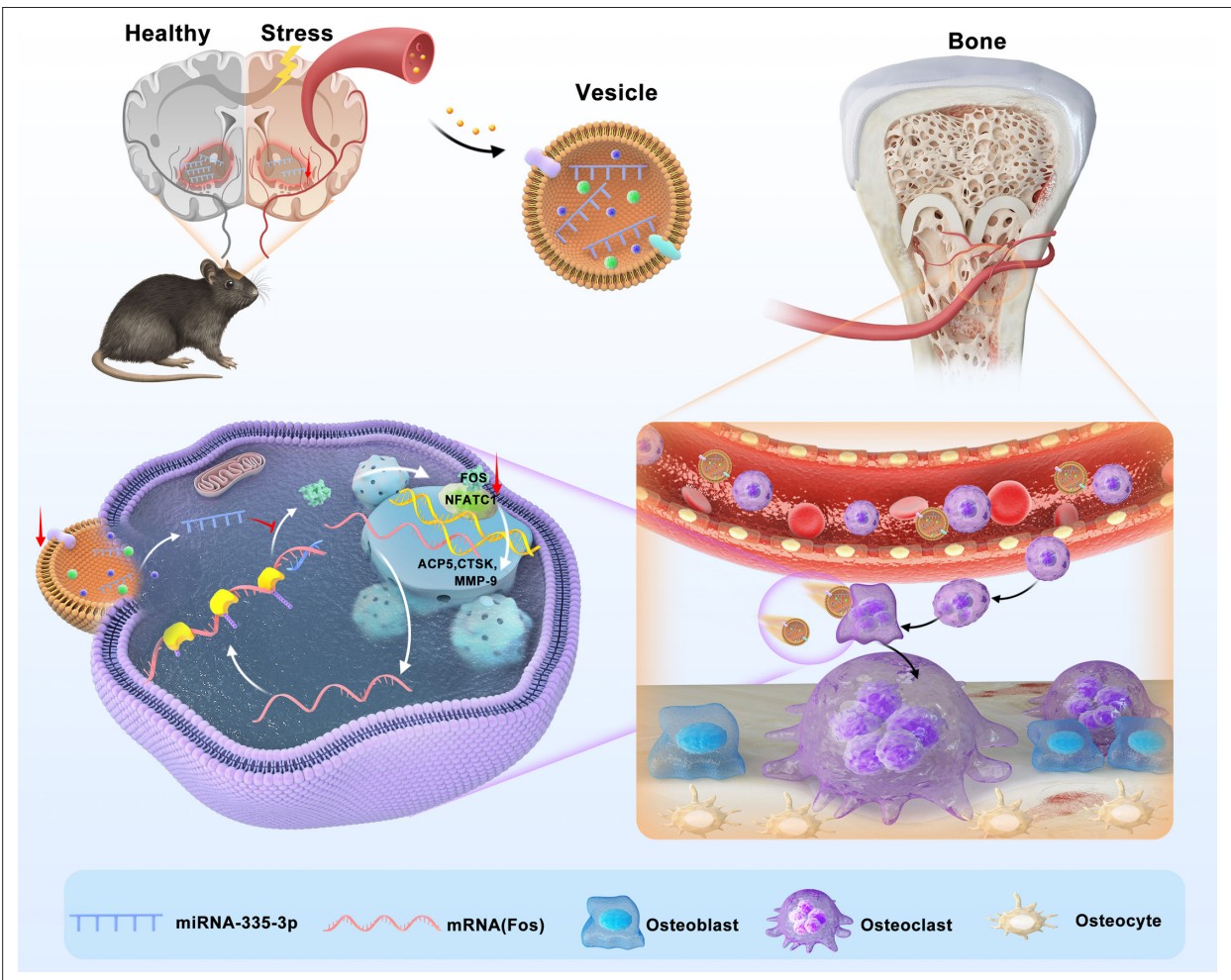

**Figure 7.** Schematic model of miR-335-3p as a regulator of osteoclast in psychological stress-induced osteoporosis. This study demonstrates that miR-335-3p levels in nucleus ambiguous (NAC), serum, and bone tissue are decreased under psychological stress, which reduces its targeting of Fos. These promote Fos translation and binding to NFATC1, increasing osteoclast activity and ultimately leading to bone loss.

miR-335-3p is one of the maturation products after shear processing of its precursor mir-335. The other is miR-335-5p, also previously known as miR-335. They may be able to synergistically regulate the progression of diseases (*Wilson et al., 2023*). Previous psychiatric disease-related studies have mainly focused on miR-335. It plays a vital role in the development of depression, both in the brain and serum (*Bocchio-Chiavetto et al., 2013*; *Fan et al., 2022*; *Li et al., 2015*; *Smalheiser et al., 2012*). Therefore, miR-335-3p, which is also present in brain-derived EVs (*Tian et al., 2022*), could be another significant miRNA involved in psychiatric disorders. At the same time, miR-335-3p also plays a crucial role in the skeletal system. One study reported a significant downregulation of miR-335-3p in the mouse bone tissue of the hindlimb unloading osteoporosis model (*Huai et al., 2020*). Our study further revealed the mechanism of miR-335-3p: it targets *Fos* so as to influence the activity of FOS-NFATC1 signaling pathway. As a result, the downregulation of miR-335-3p under chronic psychological stress promotes FOS expression and osteoclast activity, leading to osteoporosis.

In this study, we found that the osteogenic activity of CUMS mice was increased, although the bone mass was lost. This was in contrast to most other osteoporosis models caused by abnormal hormonal disturbances, including ovariectomized model (*Yang et al., 2022*), orchidectomized model (*Souza et al., 2024*), and dexamethasone-induced osteoporosis mouse model (*Rai et al., 2022*). However, the estrogenic, androgenic, and glucocorticoid alterations caused by the above modeling, respectively, are dominated by a single type of hormonal change. On the contrary, the CUMS has a more intricate situation, which involves multiple hormones and neurotransmitter changes simultaneously (*Gong et al., 2024*). Besides, *Foertsch et al., 2017* also used 7-week-old mice and established CUMS model, getting results that were in accordance with our finding of slightly upregulated mineral apposition rate (MAR). Otherwise, this difference might also indicate that some factors other than the well-known hormones involved collectively in psychological stress-induced bone loss, such as miR-335-3p that we demonstrated in this study. However, Yirmiya's study, which exposed 12-week-old mice to CUMS, found downregulation of bone formation (*Yirmiya et al., 2006*). One of the probable reasons might be that there is a self-compensatory response to bone loss triggered by chronic stress, but this self-compensatory ability of bone metabolism will weaken with age (*Little-Letsinger et al., 2022*). Nevertheless, all the studies mentioned above consistently testified that osteoclast activity was downregulated under psychological stress, assuming a more significant role of osteoclast during the pathogenesis of psychological stress-induced osteoporosis.

However, although well verified, the consistency of the trend of miR-335-3p in the NAC, serum, and femur, respectively, suggested miR-335-3p may be derived from NAC and targets bone via the blood circulation system. Further direct validation is still needed, such as using miRNA tracer mice to demonstrate the importance of brain-derived miRNA. Meanwhile, the potential impact of exogenous miR-335-3p on other bone cells, such as osteoblasts and mesenchymal stem cells, remains to be investigated. Moreover, in addition to the well-demonstrated protective effect of miR-335-3p on bone mass, effective and precisely targeted drugs need to be designed considering further therapeutic strategies for psychological stress-induced osteoporosis. Nevertheless, our study confirmed that miR-335-3p may work as a protective effector for bone metabolism during normal conditions and is a key regulator during the pathogenesis of psychological stress-induced osteoporosis.

In conclusion, our study suggests that enhanced osteoclast activity is a major cause of psychological stress-induced osteoporosis. It was found that miR-335-3p regulates osteoclast activity and protects bone metabolic balance by suppressing FOS expression and FOS-NAFCT1 signaling pathway. However, this protective effect is diminished by chronic psychological stress. These findings provide new insights into the mechanisms of osteoporosis in patients with chronic psychological stress and may aid in the development of new therapeutic strategies.

## Materials and methods
### Mice
Male C57BL/6J mice, aged 6 weeks, purchased from Shanghai Model Organisms Center, Inc, were raised at the Hubei Laboratory Animal Center, Tongji University, with a 12 hr light/12 hr dark cycle, a room temperature of 22–24°, humidity of 45–55%, and free access to food and water. All experimental procedures were approved by and performed in accordance with the standards of the Animal Welfare Committees of Tongji University in Shanghai, China (number: TJ-HB-LAC-2023-39).

## Establishment of CUMS mouse models

After 1 week of acclimation, 18 mice were divided into the control group and CUMS model group. CUMS was carried out as described previously (*Antoniuk et al., 2019*; *Willner, 2017*) with slight modifications, and the stimulations included 24 hr food deprivation, 24 hr water deprivation, 1 min tail pinch, 7 hr 45° cage tilt, 24 hr wet cage, 5 min forced swimming, 5 hr physical restraint, and 24 hr day/night reversal. These stressors were randomized to CUMS mice and continued for 6 weeks, while control mice were handled daily in the housing room. Body weight was monitored once every week at 15:30-16:30. *Figure 1A* shows the schedule of this study.

## Validation of the function of miR-335-3p in vivo

The mice were randomly divided into four groups: (1) agomir NC, (2) agomir-miR-335-3p, (3) antagomir NC, and (4) antagomir-miR-335-3p groups. Mice in the agomir NC group and agomir-miR-335-3p group were treated with negative control agomir (5 nM) and agomir (5 nM) (QianMo Biotechnology Company, Shanghai, China) by tail vein injection every 5 days. Mice in the antagomir NC group and antagomir-miR-335-3p (QianMo Biotechnology Company) group were treated with negative control antagomir (50 nM) and antagomir (50 nM) at the same time interval. The injection lasted 8 weeks.

## Behavior analysis

All behavioral tests were done in a dimly lit and quiet room. Animals were acclimatized in the room for at least 2 hr before formal testing. The testing period did not span the diurnal period of the mice. The behavioral performance was monitored and recorded using a digital camera connected to a computer running the Tracking Master software.

## Open-field test (OFT)

The mice were placed in a white open field (40 cm long, 40 cm wide, and 40 cm high). The total horizontal distance traveled in 10 min, as well as the time and distance traveled in the center area (20 cm * 20 cm), was recorded and calculated. 75% alcohol was used to clean the box during the tests of different mice.

## Tail suspension test (TST)

The mice were suspended individually, with one end of an adhesive tape fixed about 1 cm from the tip of the tail, and the other end stocked to the TST apparatus. The mice were held at a 25–30 cm distance between the tip of their nose and the ground. Their behavior was monitored for 6 min, and the immobility time in the latter 4 min was calculated.

## Sucrose preference test (SPT)

Mice were single housed and provided with two identical bottles of drinking water. First, mice were acclimatized to two bottles of normal drinking water for 24 hr. Then, they were given one bottle of normal water and one bottle of 1% sucrose water for the next 24 hr and switched the positions of the two water bottles at the 12th hour. Mice were water and food deprived for 24 hr before the test. During the test, they had free access to normal water and 1% sucrose water, and the positions of the bottles were switched in the middle of the test. The liquid consumption in each bottle over 12 hr was measured, and the sucrose preference index was calculated by dividing the consumption of sucrose solution by the total amount of fluid (water +sucrose).

## Serum ELISA

The concentrations of CORT, NE, BGP, TRAP, CTSK, and PTH in serum were measured using ELISA kits (Jiangsu MEIMIAN Industrial Co Ltd, Yancheng, China).

## Serum biochemistry

The concentrations of CA and P in serum were measured using biochemical kits (Jiangsu MEIMIAN Industrial Co Ltd).

## Micro-CT analysis

The left femur was dissected from each group of mice, fixed in 4% paraformaldehyde (PFA) for 48 hr, and stored in 0.4% PFA until scanning.

Micro-CT analysis (micro-CT 50, Scanco Medical, Zurich, Switzerland) was performed at a voxel size of 14 µm as described previously (*Dempster et al., 2013*; *Parfitt et al., 1987*; *Kim et al., 2021a*). Furthermore, 100 slices of trabecular bone underneath the growth plate (1.4 mm), and 50 slices of the cortex bone area (0.7 mm) were reconstructed for the statistical analysis. Sigma =1.2, supports =2, and threshold =200 were used to calculate the following parameters: bone volume (BV), bone volume over tissue volume (BV/TV), bone mineral density (BMD), trabecular separation (Tb. Sp), trabecular number (Tb.N), and trabecular thickness (Tb.Th).

## Histological analysis

The right femur samples were fixed with 4% PFA for 48 hr, decalcified in 0.5 mol/L ethylene diamine tetraacetic acid (pH 7.4) for 3 weeks, and paraffin-embedded. Next, the samples were sectioned into 4-um-thick slices. H&E (Beyotime, China; C0105S) was conducted to evaluate histological morphology. Osteogenic activities were detected by Masson staining (Servicebio, China; G1006) and immunohistochemical staining for OSX (Abcam, Cambridge, MA; ab22552) and OCN (Affinity, China; DF12303). For double fluorochrome labeling, a total of two intraperitoneal injections of calcein (5 mg/kg; Sigma-Aldrich, St. Louis, MO) were given at 7-day intervals at the end of CUMS modeling, and MAR was calculated using ImageJ. Osteoclastic activities were detected by immunohistochemical staining for MMP9 (Abclonal, China; A0289) and TRAP staining (Sigma-Aldrich; 387A). The number of osteoclasts (N. OCs/BS/mm) and the TRAP-positive surface of osteoclast (OC.S/B) was calculated using ImageJ. At the level of mechanism validation, immunohistochemical staining for FOS (Cell Signaling, USA; #2250), CTSK (Santa Cruz Biotechnology, CA; sc-48353), NFATC1 (Santa Cruz Biotechnology; sc-7294), and HOCHEST (Thermo Fisher, Waltham; H3570) was used to observe the location and abundance of protein expression.

## Quantitative RT-PCR analysis

Total RNA was extracted from tissues or cells using RNAiso Plus (Takara Biotechnology, Japan; 9109) according to the manufacturer's guidelines. RNA concentration was assessed with NanoDrop One. For mRNAs, cDNA was synthesized using PrimeScript RT Master Mix (Takara Biotechnology; RR036A), and a reaction mix was prepared using SYBR green master mix (Yeasen, Shanghai, China; 11201ES08). For miRNAs, miDETECT A Track miRNA qRT-PCR Starter Kit (RiboBio, Guangzhou, China; C10712-1) was used. The expression levels of the target gene were calculated using the comparative Ct ($2^{-\Delta\Delta CT}$) method with GAPDH or U6 for normalization, respectively, and using CFX Opus 96 (Bio-Rad Laboratories, Hercules, CA). The amplifications of the miRNAs were carried out using miR-335-3p-F, miR-133a-3p-F, miR-1298-5p-F, miR-144-5p-F, miR-1b-5p-F, miR-582-3p-F, miR-141-3p-F, and U6-F as forward primer and unified reverse primer as a reverse primer to amplify miR-335-3p, miR-133a-3p, miR-193b-3p, miR-1298-5p, miR-144-5p, miR-1b-5p, miR-582-3p, miR-141-3p, and U6, respectively. The primer sequences were as follows:

| Gene | 5'- 3' |
| --- | --- |
| mouse-*Opg*-F | ACCCAGAAACTGGTCATCAGC |
| mouse-*Opg*-R | CTGCAATACACACACTCATCACT |
| mouse-*Rankl*-F | CAGCATCGCTCTGTTCCTGTA |
| mouse-*Rankl*-R | CTGCGTTTTCATGGAGTCTCA |
| mouse-*Nfatc1*-F | ATGCGAGCCATCATCGA |
| mouse-*Nfatc1*-R | GGGATGTGAACTCGGAAGAC |
| Mouse-*Fos*-F | CGGGTTTCAACGCCGACTA |
| Mouse-*Fos*-R | TTGGCACTAGAGACGGACAGA |
| mouse-*Acp5*-F | GACAAGAGGTTCCAGGAGACC |

*Continued on next page*

*Continued*

| Gene | 5'- 3' |
| --- | --- |
| mouse-*Acp5*-R | GGGCTGGGGAAGTTCCAG |
| mouse-*Calcr*-F | CGTTCTTTATTACCTGGCTCTTGTG |
| mouse-*Calcr*-R | TCTGGCAGCTAAGGTTCTTGAAA |
| mouse-*Mmp9*-F | GGAACTCACACGACATCTTCCA |
| mouse-*Mmp9*-R | GAAACTCACACGCCAGAAGAATTT |
| mouse-*Ca2*-F | GCTGCAGAGCTTCACTTGGT |
| mouse-*Ca2*-R | AAACAGCCAATCCATCCGGT |
| mouse-*Oscar*-F | TGGTCATCAGTTTCGAAGGTTCT |
| mouse-*Oscar*-R | CAGCCCCAAACGGATGAG |
| mouse-*Dcstamp*-F | TGTATCGGCTCATCTCCTCCAT |
| mouse-*Dcstamp*-R | GACTCCTTGGGTTCCTTGCTT |
| mouse-*Clcn7*-F | AGCCTGGACTATGACAACAGC |
| mouse-*Clcn7*-R | GGAAAGCCGTGTGGTTGATT |
| mouse-*Ctsk*-F | GAAGCAGTATAACAGCAAGGTGGAT |
| mouse-*Ctsk*-F | TGTCTCCCAAGTGGTTCATGG |
| mouse-*Sp7*-F | ATGGCGTCCTCTCTGCTTG |
| mouse-*Sp7*-R | TGAAAGGTCAGCGTATGGCTT |
| mouse-*Ocn*-F | CTGACCTCACAGATgCCAAGC |
| mouse-*Ocn*-R | TGGTCTGATAGCTCGTCACAAG |
| mouse-*Opn*-F | AGCAAGAAACTCTTCCAAGCAA |
| mouse-*Opn*-R | GTGAGATTCGTCAGATTCATCCG |
| mouse-*Dmp1*-F | CATTCTCCTTGTGTTCCTTTGGG |
| mouse-*Dmp1*-R | TGTGGTCACTATTTGCCTGTc |
| mouse-*Runx2*-F | CCTTTACCTACACCCCGCCA |
| mouse-*Runx2*-R | GGATGCTGACGAAGTACCAT |
| mouse-*Alpl*-F | CCAACTCTTTTGTGCCAGAGA |
| mouse-*Alpl*-R | GGCTACATTGGTGTTGAGCTTTT |
| mouse-*Gapdh*-F | AGGTCGGTGTGAACGGATTTG |
| mouse-*Gapdh*-R | GGGGTCGTTGATGGCAACA |
| mmu-miR-335-3p-F | UUUUUCAUUAUUGCUCCUGACC |
| mmu-miR-1298-5p-F | UUCAUUCGGCUGUCCAGAUGUA |
| mmu-miR-144-5p-F | GGAUAUCAUCAUAUACUGUAAGU |
| mmu-miR-141-3p-F | UAACACUGUCUGGUAAAGAUGG |
| mmu-miR-133a-3p.1-F | UUGGUCCCCUUCAACCAGCUG |
| mmu-miR-193b-3p-F | AACUGGCCCACAAAGUCCCGCU |
| mmu-miR-1b-5p-F | TGGTTTTTGGGGTACATACTTCTTTAC |
| mmu-miR-582-3p-F | UAACCUGUUGAACAACUGAAC |

## microRNA sequencing (sRNA-seq)

Total RNA was extracted from M-exos and HM-exos using the Trizol reagent (Takara Biotechnology; 9109), and RNA integrity was assessed using a 2100 bioanalyzer (Agilent Technologies, CA). An Illumina TruSeq Small RNA kit (Illumina, San Diego, CA) was used to construct the library, and a high-throughput sequencing platform was used to sequence the enriched 18–32 nt small RNA fragments. The differentially expressed miRNAs in the two groups of exosomes were screened (p-value<0.05), miRDB (http://www.mirdb.org/) and DIANA (https://dianalab.e-ce.uth.gr/) databases were used to predict the target genes of the miRNAs, and KEGG enrichment analysis was performed for the identified target genes.

## 3'UTR luciferase reporter assay

A fragment of the Fos 3'UTR containing wildtype (WT) or mutant (Mut) predicted two binding sites from TargetScan (http://www.targetscan.org) database for miR-335-3p was inserted into the pMIR-REPORT vector. HEK293 cells were cotransfected with either Fos-WT or Fos-Mut vector and either miR-335-3p mimic or miRNA mimic negative control (mimic NC) using Lipofectamine 2000 (Invitrogen, USA; 11668019). Luciferase activity was determined at 48 hr post-transfection using the Dual-Glo Luciferase Assay System (Promega, Madison, WI; E2940) following the manufacturer's instructions. Data were normalized by dividing firefly luciferase activity with that of Renilla luciferase enzyme activity.

## Cell culture and transfection

The Raw 264.7 cells (RRID:CVCL_0493; gifts from Dr. Shengbing Yang, Shanghai Key Laboratory of Orthopaedic Implants, Department of Orthopaedic Surgery, Shanghai Ninth People's Hospital, Shanghai Jiao Tong University School of Medicine, Shanghai, 200125, China) were authenticated through STR profiling and tested negative for mycoplasma contamination. They were used for osteoclastic differentiation induction via RANKL (R&D Systems, USA; 462-TEC) (50 nM). The entire inducing process lasted 4 days, and medium refreshed every 3 days. In addition, Raw264.7 cells were transfected with miR-335-3p mimic, miR-335-3p inhibitor, and NC synthesized by RiboBio via Fugene HD (Promega; E2311) when cells' confluency reached 50%. For the pit formation assay, RAW264.7-derived osteoclasts were assayed on bone slices (Boneslices, Denmark), as previously described (*Luo et al., 2016*). Pits were stained with toluidine blue, and pit area was examined using ImageJ.

## Statistical analysis

Data are represented as the mean ± SD. Differences between two groups were determined by Student's *t*-test. Differences among three groups were determined by one-way ANOVA with Tukey's post hoc test. All experiments were repeated at least three times. A value of p<0.05 was considered significant.

## Acknowledgements

The authors thank the National Natural Science Foundation of China (81873709; SL), Shanghai Health Committee (20204Y0278; SL), National Natural Science Foundation of China (8227032547; WL), and Natural Science Foundation of Shanghai (20ZR1463000; WL).

## Additional information

### Funding

| Funder | Grant reference number | Author |
| --- | --- | --- |
| National Natural Science Foundation of China | 81873709 | Shuxian Lin |
| Shanghai Health Committee | 20204Y0278 | Shuxian Lin |

| Funder | Grant reference number | Author |
|--------|------------------------|--------|
| National Natural Science Foundation of China | 8227032547 | Weicai Liu |
| Natural Science Foundation of Shanghai Municipality | 20ZR1463000 | Weicai Liu |

The funders had no role in study design, data collection and interpretation, or the decision to submit the work for publication.

## Author contributions

Jiayao Zhang, Data curation, Investigation, Methodology, Writing - original draft, Writing – review and editing; Juan Li, Data curation, Formal analysis, Validation, Investigation; Jiehong Huang, Xuerui Xiang, Ruoyu Li, Software, Formal analysis, Validation, Methodology; Yun Zhai, Data curation, Methodology; Shuxian Lin, Weicai Liu, Conceptualization, Supervision, Funding acquisition, Writing – review and editing

## Author ORCIDs

Jiayao Zhang  https://orcid.org/0000-0003-2029-6665
Shuxian Lin  http://orcid.org/0000-0001-6944-5604
Weicai Liu  https://orcid.org/0000-0002-7709-6771

## Ethics

This study was performed in strict accordance with the recommendations in the Guide for the Care and Use of Laboratory Animals of the National Institutes of Health. All experimental procedures were approved by and performed in accordance with the standards of the Animal Welfare Committees of Tongji University in Shanghai, China (Number: TJ-HB-LAC-2023-39).

Reviewer #1 (Public review): https://doi.org/10.7554/eLife.95944.3.sa1
Reviewer #2 (Public review): https://doi.org/10.7554/eLife.95944.3.sa2
Author response https://doi.org/10.7554/eLife.95944.3.sa3

# Additional files

## Supplementary files

MDAR checklist

## Data availability

Sequencing data have been deposited in GEO under accession codes GSE253504. All other data generated or analysed during this study are included in the manuscript and supporting file.

The following dataset was generated:

| Author(s) | Year | Dataset title | Dataset URL | Database and Identifier |
|-----------|------|---------------|-------------|-------------------------|
| Zhang J, Liu W, Lin S, Li J, Zhai Y | 2024 | Psychological stress disturbs bone metabolism via miR-335-3p/Fos signaling in osteoclast | https://www.ncbi.nlm.nih.gov/geo/query/acc.cgi?acc=GSE253504 | NCBI Gene Expression Omnibus, GSE253504 |

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
