## [Editor Report · eLife Assessment]

The article presents **valuable** findings of bone remodeling under chronic unpredictable mild stress (CUMS). This is an interesting work on mental stress on bone health and osteoporosis, and the authors offer **solid** evidence of decreased bone mass mediated by miR-335-3p/Fos signaling in osteoclasts that are involved in the induction of bone loss caused by CUMS. This revised version provides new data that improved the quality of the article and addressed the reviewers' concerns.

---

## [Referee Report · Reviewer #1 (Public review)]

I have reviewed the manuscript "Psychological stress disturbs bone metabolism via miR-335-3p/Fos signaling in osteoclast" with interest. The described findings are relevant and useful for daily practice in periodontology. The paper is concise, professionally written, and easy to read. In this study, Jiayao et al. revealed the role of miR-335-3p in psychological stress-induced osteoporosis. CUMS mice were constructed to observe the femur phenotype, osteoclasts were identified as the main research object, and miRNA-seq was used to find the key miRNAs linking the brain and peripheral tissues. This study showed that miR-335-3p expression was simultaneously reduced in murine NAC, serum, and bone under psychological stress. The miR-335-3p/Fos/NFATC1 signaling pathway was validated in osteoclasts to reveal the potential mechanism of enhanced osteoclast activity under psychological stress. This study, from a new perspective of miRNAs, indicates a possible cause of disturbed bone metabolism due to psychological stress and may suggest a new approach to treating osteoporosis.

---

## [Referee Report · Reviewer #2 (Public review)]

Zhang et al. established chronic unpredictable mild stress (CUMS) mouse model, which displayed osteoporosis phenotype, suggesting a potential correlation between psychological stress and bone metabolism. They found that miRNA candidate miR-335-3p is downregulated in the long bone of CUMS mice through microRNA sequencing experiments and qRT-PCR. They further demonstrated that miR-335-3p attenuates osteoclast activity via inhibiting Fos signaling, which can induce NFATC1 expression and regulate osteoclast activity.

My concerns have been addressed. And the quality of the manuscript is improved significantly.

---

## [Author Response]

The following is the authors’ response to the original reviews.

**Public Reviews:**

**Reviewer #1 (Public Review):**
I have reviewed, with interest, the manuscript "Psychological stress disturbs bone metabolism via miR-335-3p/Fos signaling in osteoclast". The described findings are relevant and useful for daily practice in periodontology. The paper is concise, professionally written, and easy to read. In this study, Jiayao et al. revealed the role of miR-335-3p in psychological stress-induced osteoporosis. CUMS mice were constructed to observe the femur phenotype, osteoclasts were identified as the primary research object, and miRNA-seq was used to find the key miRNAs linking the brain and peripheral tissues. This study showed that the expression of miR-335-3p was simultaneously reduced in mice's NAC, serum, and bone under psychological stress. The miR-335-3p/Fos/NFATC1 signaling pathway was validated in osteoclasts to reveal the potential mechanism of enhanced osteoclast activity under psychological stress. From a new perspective of miRNAs, this study indicates a possible cause of disturbed bone metabolism due to psychological stress and may suggest a new approach to treating osteoporosis.

We thank this reviewer for the instructive suggestions and encouragement.

**Reviewer #2 (Public Review):**
Zhang et al. established chronic unpredictable mild stress (CUMS) mouse model, which displayed osteoporosis phenotype, suggesting a potential correlation between psychological stress and bone metabolism. They found that miRNA candidate miR-335-3p is downregulated in the long bone of CUMS mice through microRNA sequencing and qRT-PCR experiments. They further demonstrated that miR-335-3p attenuates osteoclast activity via inhibiting Fos signaling, which can induce NFATC1 expression and regulate osteoclast activity.Strengths:The authors established CUMS mouse model and confirmed the osteoporosis phenotype through careful characterization of bone and analysis of osteoclast activity. They performed microRNA sequencing to identify the miRNA candidate regulating the bone loss in the CUMS mouse model. They also validated the expression of miR-335-3p and interfered with the function of miR-335-3p through an in vitro assay. Overall, the findings from this study provide important hints for the correlation between psychological stress and bone metabolism.

We thank this reviewer for the comprehensive summary and positive comment on our work.

Weakness:The data provided by the authors are preliminary, especially the mechanistic insight, which needs to be enhanced. The authors have shown that miR-335-3p expression was altered in the CUMS mouse model and the change of its expression regulated osteoclast activity. The validation should be conducted in vivo, and the mechanism behind this should be investigated further.

We thank the reviewer’s important insight on the need for further in vivo validation of the role of miR-335-3p. Therefore, we designed and produced Antagomir-335-3p (antagonist) and Agomir-335-3p (agonist). Then, we injected them into the body through the tail vein for about 2 months and observed the bone phenotype in each group of mice. The results suggested that the decrease of miR-335-3p in vivo could lead to bone loss, which was consistent with our in vitro validation results (Figure 5H-I).

**Reviewing Editor:**
Method(1) Bone histomorphometric analysis following ASBMR's guidelines Bone histomorphometric analysis of bone formation and bone resorption: The authors should follow ASBMR's guidelines for bone histomorphometry (PMCID: PMC3672237 and PMID: 3455637) to perform standard analyses of histomorphometry, rather than selected areas. They should also clearly describe a software used and define the areas analyzed.

We carefully re-analyzed bone histomorphometry according to ASBMR guidelines and combine this with our own understanding. At the same time, we improved the description of micro-CT and histological analysis in the method. If there is still any lack of standardization, we would be grateful for any constructive suggestions to improve this.

(2) Osteoclast cultures require nuclear staining to demonstrate multinucleated Trap positive cells.

We used the RAW264.7, a mouse macrophage-like cell line, for in vitro culture and induced its differentiation towards osteoclasts. Successfully induced osteoclasts showed enlarged cytoplasm and multinucleated fusion. Tartrate-resistant acid phosphatase (Trap) is the signature enzyme of osteoclasts. It can bind to the chromogen to exhibit a mauve color, based on the principle of azo-coupled immunohistochemistry. At the same time, small and rounded nuclei fused show a lighter color (author response image 1, yellow arrows). We attempted to stain the nuclei with hematoxylin based on this. However, it was unable to further distinguish the contours of the nuclei clearly due to the similar color to the Trap positive signals. Besides, many other scholars have assessed osteoclast activity in vitro experiments based solely on the results of Trap staining (area and number) (Cheng et al., 2022; Li et al., 2019; Ma et al., 2021; Zhong et al., 2023). Nevertheless, in the immunofluorescence staining of osteoclasts, the nuclei were labeled using a Hochest antibody to reflect the multinucleated fusion of osteoclasts (Figure 5G).

(3) Osteoclast pit assays should be carried out to necessarily demonstrate the change of osteoclast resorption ability caused by miR-335-3p.

We added osteoclast pit assays to validate the role of miR-335-3p on osteoclast resorptive capacity (Figure 5D-E).

(4) Serum ELISA assay should be done to examine the global change of bone remodeling in the CUMS mice to assess bone formation and bone resorption that will support their claim.

We performed additional tests on serum concentrations of R-hydroxy glutamic acid protein (BGP), TRAP, Cathepsin K (CTSK), parathyroid hormone (PTH), calcium (CA), phosphate (P) in control and CUMS mice, which could better reflect the global change of bone remodeling in the CUMS mice (Figure 3— figure supplement 1).

(5) miR-RNA-seq: A labeled volcano plot should be used to replace the present one to show significant changes in differential gene expression.

We appreciate this great suggestion. We replaced the volcano plot that showed significant changes in differential gene expression (Figure 4B). We also uploaded the raw data to the GEO database (GSE253504), making the results clearer and more accessible.

DiscussionThe authors should discuss previous works on the influences of hormones from the brain on chronic stress-induced bone loss and an association of these influences with their findings.

The discussion on the relationship between the bone metabolism regulation of both hormones and miR-335-3p in psychological stress was added in the second and fifth paragraphs of the discussion. To conclude, on the one hand, brain-derived and blood-transported miR-335-3p regulate bone metabolism synergistically. On the other hand, it exerted a more direct influence on bone under psychological stress.

LanguageThe language of the MS should be improved.

The manuscript has been carefully edited by a professional proofreader.

**Reviewer #1 (Recommendations For The Authors):**
(1) Figure 1F: The exact meaning of the Waveform Graph shown at left needs to be clarified for the not-so-experienced reader.

We added the more detailed meaning of the Waveform Graph in figure legends (Figure legend 1F).

(2) Is the concomitant increase in osteogenic and osteoblastic activity in this study consistent with that seen in similar disease studies? This could be added to the discussion.

In the fifth paragraph of the discussion section, we present the alterations of osteogenic and osteoblastic activity observed in other studies that are similar to ours. We also had a detailed discussion based on these observations.

(3) Figure 6A: Please highlight the key information to visualize the potential linkage among miR-335-3p, Fos, and osteoclast.

We highlighted the crucial linkage among miR-335-3p, Fos, and osteoclast with red arrows (Figure 6A)

1. Figure 6E: The specific area of the selected comparison needs to be clarified. Please add white dotted lines and lettering T (trabecular bone) and GP (growth plate) for the not-so-experienced reader. This will provide some orientation.

We used white dotted lines as well as letters to label the tissue in immunofluorescence staining images (Figure 6E).

(5) Line 350: "NAC derived and blood-trans, Ported miR-335-3p". There is a grammatical error. Please conduct general proofreading of the text and writing style.

Thank you for pointing this out. We have corrected this grammatical error, and we also checked the full text to correct similar errors.

**Reviewer #2 (Recommendations For The Authors):**
(1) miR-335-3p was downregulated in the femur in the CUMS mice. The possible mechanism for this outcome should be further discussed. In Figure 4B, the Volcano plot showed that only a few miRNA were differentially expressed between the control and CUMS mice. How do the authors explain this?

The chronic unpredictable mild stress (CUMS) model was constructed using normal mice. As the name of the model suggests, the stimulus is mild and does not cause developmental damage or teratogenic effects in mice. Conversely, CUMS has the potential to result in the chronic pathological conditions. Besides, in miRNA sequencing results from other tissues with similar models to ours, the number of differential miRNAs is also around a few dozen (Ma et al., 2019).

(2) The authors have demonstrated that miR-335-3p inhibits osteoclast differentiation based on an in vitro assay in Figure 5; however, an in vivo experiment is required to provide more solid evidence.

We strongly agree that in vivo experimental validation would bring more convincing results to this study. Therefore, we designed and produced Antagomir-335-3p (antagonist) and Agomir-335-3p (agonist), which were injected into mice via the tail vein every five days. Samples were collected at one and two months following the injection. We found that sustained two-month injections of antagomir could significantly lead to bone loss in mice (Figure 5H-I), which is consistent with our in vitro validation results.

However, the Agomir-miR-335-3p group did not exhibit a notable enhancement of bone mass. This may be attributed to the fact that the 11-week-old normal mice selected for this study were in their prime and did not have strong osteoclastic activity in vivo. Therefore, the osteoclastic inhibition of Agomir-335-3p could not be demonstrated.

In addition, no significant difference was seen one month after the injection. The main reason may be that the time is too short. On the one hand, the drug we injected was RNA preparation. They lacked stability resulting in poor delivery efficiency, which took some time to take effect. On the other hand, bone remodeling is also a time-consuming process.

(3) FOS and NFATC1 should be expressed in the nuclei of the cells, therefore, the quality of the images needs to be improved.

NFATC1 is a T-cell-activating nuclear factor that is activated in the nucleus to regulate the transcription of a variety of osteoclast-related genes, including ACP5, MMP9, etc. FOS could bind and interact with NFATC1, resulting in nuclear translocation and transcription activated. This could promote the differentiation and maturation of osteoclasts. They are both synthesized and processed in the cytoplasm and eventually enter the nucleus to perform their functions. Therefore, they are expressed in both the nucleus and the cytoplasm (Deng et al., 2022; Hounoki et al., 2008; Li et al., 2022).

In Figure 5G, we labeled cell nuclei with HOCHEST antibody with blue fluorescence, and more co-localized signals of nuclei (blue), FOS (red), and NFATC1 (green) were seen in the Inhibitor-miR-335-3p group, whereas the opposite result was observed in the Mimic-miR-335-3p group. These results indicated that inhibited miR-335-3p could promote osteoclast differentiation in vitro.

(4) The expression of FOS was elevated in CUMS group in Figure 6E; however, its mRNA level was unchanged, as shown in Figure 6 supplement; what is the explanation for this? How do the authors claim FOS is the downstream target if its mRNA expression is not impacted by CUMS?

The results demonstrated that miR-335-3p targeted binding to the mRNA of *Fos* did not result in mRNA degradation. Instead, this binding interferes with the protein translation process, which ultimately leads to the reduction of FOS protein.

(5) What would be the bone phenotype if a FOS inhibitor was injected into the control and CUMS mice? It is important to examine FOS function through an in vivo context.

The regulatory role of FOS for osteoclasts has been validated in numerous articles, both in vivo and in vitro(Aikawa et al., 2008; Cao et al., 2023; Cheng et al., 2022). For example, Aikawa et al. designed a small-molecule inhibitor of c-Fos/activator protein-1 (AP-1) using three-dimensional (3D) pharmacophore modeling, which helped verify the effect of FOS on osteoclasts *in vivo*(Aikawa et al., 2008).

We also strongly agree that *in vivo* injection of inhibitors of FOS, especially in CUMS mice, could further substantiate the role of miR-335-3p in osteoclasts under psychological stress. However, the study was constrained by the unavailability of commercially viable, efficacious small molecule inhibitors of FOS. In the future, we plan to design more precise therapeutic targets for psychological stress induced osteoporosis based on existing research ideas.

Reference

Aikawa, Y., Morimoto, K., Yamamoto, T., Chaki, H., Hashiramoto, A., Narita, H., Hirono, S., & Shiozawa, S. (2008). Treatment of arthritis with a selective inhibitor of c-Fos/activator protein-1. Nature Biotechnology, 26(7), 817-823. https://doi.org/10.1038/nbt1412

Cao, Z., Niu, X. B., Wang, M. H., Yu, S. W., Wang, M. K., Mu, S. L., Liu, C., & Wang, Y. X. (2023, Nov). Anemoside B4 attenuates RANKL-induced osteoclastogenesis by upregulating Nrf2 and dampens ovariectomy-induced bone loss [Article]. Biomedicine & Pharmacotherapy, 167, 12, Article 115454. https://doi.org/10.1016/j.biopha.2023.115454

Cheng, X., Yin, C., Deng, Y., & Li, Z. (2022). Exogenous adenosine activates A2A adenosine receptor to inhibit RANKL-induced osteoclastogenesis via AP-1 pathway to facilitate bone repair. Molecular Biology Reports, 49(3), 2003-2014. https://doi.org/10.1007/s11033-021-07017-1

Deng, W., Ding, Z., Wang, Y., Zou, B., Zheng, J., Tan, Y., Yang, Q., Ke, M., Chen, Y., Wang, S., & Li, X. (2022). Dendrobine attenuates osteoclast differentiation through modulating ROS/NFATc1/ MMP9 pathway and prevents inflammatory bone destruction. Phytomedicine : International Journal of Phytotherapy and Phytopharmacology, 96, 153838. https://doi.org/10.1016/j.phymed.2021.153838

Hounoki, H., Sugiyama, E., Mohamed, S. G.-K., Shinoda, K., Taki, H., Abdel-Aziz, H. O., Maruyama, M., Kobayashi, M., & Miyahara, T. (2008). Activation of peroxisome proliferator-activated receptor gamma inhibits TNF-alpha-mediated osteoclast differentiation in human peripheral monocytes in part via suppression of monocyte chemoattractant protein-1 expression. Bone, 42(4), 765-774. https://doi.org/10.1016/j.bone.2007.11.016

Li, Y., Yang, C., Jia, K., Wang, J., Wang, J., Ming, R., Xu, T., Su, X., Jing, Y., Miao, Y., Liu, C., & Lin, N. (2022). Fengshi Qutong capsule ameliorates bone destruction of experimental rheumatoid arthritis by inhibiting osteoclastogenesis. Journal of Ethnopharmacology, 282, 114602. https://doi.org/10.1016/j.jep.2021.114602

Li, Z., Huang, J., Wang, F., Li, W., Wu, X., Zhao, C., Zhao, J., Wei, H., Wu, Z., Qian, M., Sun, P., He, L., Jin, Y., Tang, J., Qiu, W., Siwko, S., Liu, M., Luo, J., & Xiao, J. (2019). Dual Targeting of Bile Acid Receptor-1 (TGR5) and Farnesoid X Receptor (FXR) Prevents Estrogen-Dependent Bone Loss in Mice. Journal of Bone and Mineral Research : the Official Journal of the American Society For Bone and Mineral Research, 34(4), 765-776. https://doi.org/10.1002/jbmr.3652

Ma, K., Zhang, H., Wei, G., Dong, Z., Zhao, H., Han, X., Song, X., Zhang, H., Zong, X., Baloch, Z., & Wang, S. (2019). Identification of key genes, pathways, and miRNA/mRNA regulatory networks of CUMS-induced depression in nucleus accumbens by integrated bioinformatics analysis. Neuropsychiatric Disease and Treatment, 15, 685-700. https://doi.org/10.2147/NDT.S200264

Ma, Q., Liang, M., Wu, Y., Luo, F., Ma, Z., Dong, S., Xu, J., & Dou, C. (2021). Osteoclast-derived apoptotic bodies couple bone resorption and formation in bone remodeling. Bone Research, 9(1), 5. https://doi.org/10.1038/s41413-020-00121-1

Zhong, L., Lu, J., Fang, J., Yao, L., Yu, W., Gui, T., Duffy, M., Holdreith, N., Bautista, C. A., Huang, X., Bandyopadhyay, S., Tan, K., Chen, C., Choi, Y., Jiang, J. X., Yang, S., Tong, W., Dyment, N., & Qin, L. (2023). Csf1 from marrow adipogenic precursors is required for osteoclast formation and hematopoiesis in bone. eLife, 12. https://doi.org/10.7554/eLife.82112